# Folate Status as a Nutritional Indicator among People with Substance Use Disorder; A Prospective Cohort Study in Norway

**DOI:** 10.3390/ijerph19095754

**Published:** 2022-05-09

**Authors:** Mitra Bemanian, Jørn Henrik Vold, Ranadip Chowdhury, Christer Frode Aas, Rolf Gjestad, Kjell Arne Johansson, Lars Thore Fadnes

**Affiliations:** 1Bergen Addiction Research, Department of Addiction Medicine, Haukeland University Hospital, 5012 Bergen, Norway; jorn.henrik.vold@helse-bergen.no (J.H.V.); christer.frode.aas@helse-bergen.no (C.F.A.); kjell.johansson@uib.no (K.A.J.); lars.fadnes@uib.no (L.T.F.); 2Department of Global Public Health and Primary Care, Faculty of Medicine, University of Bergen, 5009 Bergen, Norway; 3Division of Psychiatry, Haukeland University Hospital, 5036 Bergen, Norway; rolf.gjestad@uib.no; 4Centre for Health Research and Development, Society for Applied Studies, New Delhi 110016, India; ranadip.chowdhury@sas.org.in

**Keywords:** substance use disorder, nutrition, folate, vitamin, opioid agonist therapy

## Abstract

Substance use disorder (SUD) is associated with poor nutrition. Vitamin B9, or folate, is an important micronutrient for health. The aim of this prospective longitudinal cohort study was to assess serum folate levels among people with SUD and to investigate the impact of factors related to substance use severity on folate status. Participants were recruited from outpatient clinics for opioid agonist therapy (OAT) and municipal health-care clinics for SUD in Western Norway. They were assessed annually, including blood sampling for determination of micronutrient status. Overall, 663 participants with a total of 2236 serum folate measurements were included. A linear mixed model was applied, and measures are presented as β-coefficients with 95% confidence interval (CI). Forty-eight percent (CI: 44–51) of the population had low serum folate levels (s-folate < 10 nmol/L), and 23% (CI: 20–26) were deficient (s-folate < 6.8 nmol/L) at baseline. Sixty percent (CI: 53–65) sustained their poor folate status in at least one subsequent assessment. Except for weekly use of cannabis (mean difference in serum folate [nmol/L]: −1.8, CI: −3.3, −0.25) and alcohol (1.9, CI: 0.15, 3.6), weekly use of no other substance class was associated with baseline differences in serum folate when compared to less frequent or no use. Injecting substances was associated with a reduction in serum folate over time (−1.2, CI: −2.3, −0.14), as was higher dosages of OAT medication (−1.1, CI: −2.2, −0.024). Our findings emphasize the need of addressing nutrition among people with severe SUD.

## 1. Introduction

Folate deficiency is a known complication of alcohol use disorder, but an association of folate status with other substance use disorders (SUDs) is not well-established [1,2]. Under- and malnutrition is common among people with SUD, in part due to limited food accessibility and unfavourable dietary habits [3,4]. The diet of those heavily burdened by SUD tends to be dominated by foods with a low nutritional value, in particular simple carbohydrates and sugar-sweetened snack foods and beverages [5,6,7]. This type of diet increases the risk of many micronutrient deficiencies, including that of folate deficiency [1,6]. Although there is some evidence on overall low dietary intake [8,9] and poor blood status of folate [10,11,12] among people with SUD, there is little updated literature on folate status and risk factors for folate deficiency in this population.

Folate, or vitamin B9, is an essential micronutrient that is particularly abundant in a range of fruits, legumes, and green leafy vegetables [1,13]. An adequate blood status of folate is dependent on a well-balanced diet, especially in countries where grain and cereal products are not routinely fortified with its synthetic equivalent folic acid [1,14]. On a molecular level, folate is a key cofactor in one-carbon reactions vital for cell proliferation and genomic integrity [13,15]. Folate deficiency is an established cause of megaloblastic anaemia, pregnancy complications and foetal developmental malformations [13,16]. Further, sustained low or deficient folate status is associated with an increased risk of cognitive decline and neurological disease and symptoms, depressive disorders as well as some cancers [17,18,19,20].

Opioid agonist therapy (OAT), with methadone or buprenorphine-based medication, is widely accepted as the gold standard for treatment of severe opioid dependence [21]. The OAT is a well-documented medical intervention for reducing morbidity and mortality, and improving the health-related quality of life, of individuals with opioid dependence [22,23,24]. Nevertheless, patients in OAT have a significant disease burden and increased morbidity and mortality compared to the general population [25,26,27]. The high mortality of patients in OAT is equally attributable to substance overdoses and comorbid somatic disease, including chronic infections such as hepatitis C virus, cardiovascular and pulmonary disease [28,29]. Additionally, long-term use of opioids, including opioids for OAT, is associated with adverse metabolic and endocrinologic manifestations including the development of overweight, dyslipidaemia, hypogonadism, and metabolic syndrome [30,31,32]. Efforts should be made to reduce the disease burden in this population, and this includes addressing complications related to adverse dietary habits, food accessibility and nutritional deficiencies [6].

The poor dietary and nutritional status of people with SUD, and particularly the tendency to replace nutritious foods with so-called “empty calories” in the form simple carbohydrates and dietary sugar, puts them at high risk of developing folate deficiency [6]. Little is known about the extent of folate deficiency in this population, and whether the substance use patterns, injecting substance use, or treatment with opioid agonist therapy are associated with the levels of folate. Thus, the objective of this longitudinal cohort study was to determine the serum folate status of a population with SUD visiting outpatient OAT clinics and municipality health-care clinics for SUD in Bergen and Stavanger, Norway. Further, we aimed to assess the impact of sociodemographic and clinical factors related to severity of substance use on serum folate status.

## 2. Materials and Methods

### 2.1. Study Characteristics; Design, Population, Data Collection and Study Sample

This is a prospective longitudinal cohort study nested to the multicentre INTRO-HCV and ATLAS4LAR studies [33,34]. Participants were recruited from a population of outpatients visiting OAT clinics or municipality health-care clinics for SUD in Bergen and Stavanger, Norway. Participants were assessed yearly with a specialized research nurse-led and questionnaire-based interview focused on somatic and mental health, psychosocial aspects and substance use patterns. Data were collected using the software CheckWare^®^ (CheckWare, Trondheim, Norway). Clinical data were obtained from the electronic medical record. Data collected between May 2016 and June 2020 are presented. A total of 2236 valid serum folate measurements from 663 participants were included. Each participant had a mean of 3.4 (Standard Deviation (SD): 3.6) serum folate measurements during the study period, and a total of 516 (78%) participants had at least two serum folate measurements.

### 2.2. Measuring Serum Folate; Laboratory Assays and Definitions

Venous blood samples were collected according to standard protocol and sent to the Department of Medical Biochemistry and Pharmacology at Haukeland University Hospital in Bergen and the Department of Medical Biochemistry at Stavanger University Hospital in Stavanger for analysis of serum folate concentration (both accredited by ISO-standard 15189). The former laboratory assessed folate concentration in serum samples by means of the Electrochemiluminescence Immunoassay (10% analytical variation at concentration 8.4 nmol/L) [35], whereas the latter used the Chemiluminescence Microparticle Immunoassay (12% analytical variation at concentration 9 nmol/L) [36]. Data on serum folate concentration was obtained from the electronic medical record. The unit used was nanomoles per litre (nmol/L), and values between 1.5 and 45.3 nmol/L were specified. Two separate cut-offs were used when describing folate status: folate deficiency was defined as s-folate < 6.8 nmol/L. This cut-off is set based on the risk of developing megaloblastic anaemia, and is widely used in literature when describing folate deficiency [37]. Low folate status was defined as s-folate < 10 nmol/L. This cut-off is based on metabolic manifestations of negative folate balance, namely elevated plasma homocysteine concentration [37]. This is also the cut-off which, according to local guidelines based on recommendations from the World Health Organization (WHO), warrants intervention in the form of nutritional guidance or supplementation [37,38,39].

### 2.3. Study Variables; Baseline, OAT, Clinical and Sociodemographic Factors

The baseline was defined as the serum folate measurement performed in closest proximity in time to the first health assessment for the individual. The subsequent serum folate measurements were listed chronologically and included as follow-up, and time was defined as years from baseline. The OAT was defined as receiving methadone or buprenorphine-based medication. We calculated an OAT dose ratio corresponding to the prescribed daily dose of medication divided by the mean of WHO’s recommended dose interval (90 mg for methadone and 18 mg for buprenorphine) [40]. As for the clinical factors, injecting substances was defined as having injected any substance within the prior six months. Frequent substance use was defined as using any of the following substances on a minimum weekly basis during the 12 months leading up to the annual health assessment: alcohol, cannabis, benzodiazepines, stimulants (amphetamines and cocaine) and non-OAT opioids (e.g., heroin). Hepatitis C virus (HCV) status was determined by means of a quantitative polymerase chain reaction assay, and non-zero values were defined as HCV infection. Regarding sociodemographic factors, housing conditions for the last 30 days leading up to the annual health assessment were defined as stable (living in owned or rented home or at an institution) or unstable (being homeless, living at shelter or with friends and family). Highest completed education level was classed into the following groups: primary school (7 years), middle school (10 years), high school (13 years), ≤3 years of college or university level education and >3 years of college or university level education. Age was categorized into the following groups: <30 years, 30–39 years, 40–49 years, 50–59 years and ≥60 years.

### 2.4. Statistical Analyses

The Stata/SE 16.0 software (Stata Corporation, College Station, TX, USA) was used for the generation of descriptive statistics and a linear mixed model (LMM). The SPSS version 26.0 software (IBM, Armonk, NY, USA) was used for expectation maximization imputation. The software R version 4.0.3 (R foundation for Statistical Computing, Vienna, Austria) with the package mgcv was used for the preparation of generalized additive models (GAMs) and their graphical presentations. The website Sankeymatic (http://www.sankeymatic.com/build, accessed on 1 September 2021) was used for the generation of a Sankey diagram. The threshold of statistical significance was set to *p* < 0.05 for all analyses. Nine percent of values were missing across the sociodemographic and clinical variables, including variables assessing substance use patterns. These were assumed to be missing at random and expectation maximization imputation was performed to replace them with estimated values [41]. Descriptive data are presented with total numbers and percentages, only including valid, non-imputed values. Median (Interquartile range (IQR)) serum folate levels for different subgroups are presented. The prevalence of low and deficient folate status within subgroups are presented as percentages with 95% confidence intervals (CI). The CIs for prevalence measures were estimated by means of the one proportion Z-test. The GAM models and plots were generated to visualize the nonlinear associations of substance use severity with serum folate, adjusted for gender, age and education level. In order to do this, a substance use severity score was generated based on the type, frequency and number of substances used (for details see Appendix A). A Sankey diagram was generated to display the flow between folate status categories (i.e., deficient, low and adequate folate status) between the first and second folate assessment for participants with at least two folate measurements (n = 516). A LMM was performed to estimate associations of clinical and sociodemographic factors with baseline serum folate concentration, and the impact of selected clinical factors on changes in serum folate concentration over time (including OAT dose ratio, injecting substances, and weekly use of substances). The model was random intercept fixed slope with the estimator set to restricted maximum likelihood. Time was defined as years from baseline, and predictor variables were kept constant to the value held at baseline. To estimate the contribution of individual clinical factors, partial adjusted models predicting associations of individual clinical factors with serum folate concentration (adjusted for gender and age) are presented in addition to the adjusted model including all variables.

## 3. Results

The baseline characteristics of the study sample are presented in Table 1. The mean age at baseline was 44 years (SD: 11), and 70% were males. Eighty-eight percent were patients enrolled in OAT, and of these 60% were prescribed buprenorphine-based medication and 40% were prescribed methadone. Twelve percent lived under unstable housing conditions, 47% were infected with HCV and less than five participants were human immunodeficiency virus (HIV) positive at baseline. Weekly substance use was reported by 76%, and the most common substances used on a weekly basis were cannabis (49%) and benzodiazepines (38%), followed by stimulants (26%) and alcohol (25%).

The median (IQR) serum folate concentration of the study population was 10 (11) nmol/L at baseline (Table 1). The distribution was leptokurtic (kurtosis: 4.9) and right-skewed (skewness: 1.5). Forty-eight percent (CI: 44–51) of the population had a low serum folate status and 23% (CI: 20–26) were deficient at baseline (Table 1). Low folate status was particularly prevalent among those under the age of 30 (59%, CI: 48–69), those injecting substances (55%, CI: 50–60) or using stimulants on a weekly basis (54%, CI: 46–62), and those prescribed methadone for OAT (54%, CI: 48–60). The prevalence of low folate status was similar between participants receiving buprenorphine for OAT (44%, CI: 39–49) compared to those not receiving OAT (45%, CI 34–57). Folate deficiency was however substantially more prevalent among those prescribed methadone (32%, CI: 26–38) compared to those prescribed buprenorphine (18%, CI: 14–22) or not receiving OAT (15%, CI: 9–25).

Figure 1 shows that 63% (CI: 57–69) of participants with an adequate serum folate status, 60% (CI: 53–65) of those with low levels and 43% (CI: 34–51) of those deficient at baseline sustained this status in the subsequent assessment.

Figure 1 displays the movement between folate status categories from the first (left) to the second (right) folate assessment for participants with at least two serum folate measurements (n = 516). It shows that 63% (CI: 57–69) of participants with an adequate serum folate status, 60% (CI: 53–65) of those with low levels and 43% (CI: 34–51) of those deficient at baseline sustained this status in the subsequent assessment. Definitions: Adequate (green) = s-folate > 10 nmol/L, Low (yellow) = s-folate 6.8–10 nmol/L, Deficient (red) = s-folate < 6.8 nmol/L.

Figure 2 shows that serum folate concentration was inversely associated with the substance use severity score at low to mid-range scores at baseline (scores 0–12). There were few participants with the highest substance use severity scores and thus limited precision for the highest substance use severity (7.5% had scores > 15).

In Figure 2, the graph to the left displays a GAM plot of the associations of serum folate concentrations with the substance use severity score the solid line depicts the association at various severity scores, whereas the shaded area represents the 95% confidence intervals of these associations. The plot to the right displays the distribution of substance use severity scores in the population. A substance use severity score of 0 equals no use of substances, whereas a score of 25 equals daily use of 5 substance classes (cannabis, stimulants, opioids, alcohol and benzodiazepines). Serum folate concentration was inversely associated with the substance use severity score at low to mid-range scores at baseline (scores 0–12). There were few participants with substance use severity scores >15 resulting in limited precision for these values.

Table 2 presents the results of a LMM of serum folate predicted by sociodemographic and clinical factors. Higher serum folate concentrations were found for those over the age of 50 years (mean serum folate difference: 2.4, CI: 0.32, 4.5) and sixty years (4.5, CI: 1.5, 7.4) compared to those under the age of 30 years at baseline. Weekly use of alcohol was associated with higher serum folate concentrations at baseline compared to less or no alcohol use (1.9, CI: 0.15, 3.6), whereas weekly use of cannabis was associated with lower serum folate concentrations (−1.8, CI: −3.3, −0.25) when compared to less or no use. Statistically significant negative associations were found between injecting substances and serum folate concentration over time (−1.2, CI: −2.3, −0.14), and between higher OAT dose ratios and serum folate concentration over time (−1.1, CI: −2.2, −0.024). The β-coefficients were similar between partly adjusted and fully adjusted analyses. One exception was the effect estimate of the variable for injecting substances which was significantly associated with folate in the partly adjusted model (−1.6, CI: −3.1, −0.13), while non-significant in the fully adjusted model (−1.2, CI: −2.9, 0.41). This is likely due to correlation between the variable for injecting substances and the variable for weekly use of stimulants (correlation coefficient: 0.41)

## 4. Discussion

Overall, the serum folate status of this Norwegian population of outpatients with SUD was poor. Nearly half of the population had serum folate levels below the cut-off that warrants intervention in the form of nutritional guidance or supplementation, and 23% were deficient. Additionally, poor folate status was to a large part sustained over time. Although we lack healthy controls and comparable data on the prevalence of folate deficiency in the general Norwegian population, studies from other high-income countries generally report the prevalence of folate deficiency to under five percent, in contrast to low-income regions where the prevalence is in the realms of twenty percent [42,43]. It is, however, important to note that many of the comparable high-income countries have implemented mandatory food fortification programs that include folic acid [44]. Norway has not implemented this, and the prevalence of folate deficiency is therefore feasibly much higher. Denmark is comparable to Norway in terms of demographics as well as fortification policies, and a study on a representative Danish sample showed that a third had poor serum status of folate and that lifestyle factors, in particular diet quality, were associated with this [45]. In regions where foods are not routinely fortified with folic acid, adequate intake of the folate is highly dependent on a sufficiently balanced diet rich in its primary sources including fruits, vegetables, and grain products [14,46]. The poor folate status presented in this study on people with SUD is likely indicative of an insufficient dietary intake, which in turn could reflect an overall poor dietary status—as is widely reported in literature [5,6,7,8,11]. In line with recommendations from WHO on tackling poor folate status in at-risk populations, we argue that any actions taken towards improving the folate status of people with SUD should, when feasible, be long-term and keep a broad scope on diet and nutrition [16].

Substance use was negatively associated with serum folate status. Higher severity of substance use, meaning more frequent use of multiple substances, was associated with lower serum folate concentration at baseline. Additionally, two factors related to SUD severity, namely injecting substances, and higher dosages of OAT medication, were associated with reduced serum folate concentration over time. This is consistent with other literature reporting higher rates of anthropometric and protein malnutrition among people with more intense [47] and prolonged substance use [48], and among those who inject substances or use heroin in combination with other substances [49]. Low folate levels were more frequent among those using methadone (a full agonist to opioid receptors) compared to buprenorphine (partial agonist) or no OAT. This could indicate an opioid related effect or interaction between opioids and eating behaviour but could also be due to selection of people with more severe substance use disorder more often using methadone compared to buprenorphine. As mentioned, the poor folate status in this population of outpatients with SUD is likely indicative of insufficient dietary intake, which again could reflect an overall poor nutritional status. A nutritional assessment of a comparable Norwegian population with severe substance use revealed largely monotonous and unfavourable diets high in dietary sugar and low in nutritious foods, including fruits and vegetables [6]. Similar tendencies have been reported in other studies on people with SUD [5,7,8,11]. Opioid users in particular, including patients enrolled in OAT, are often reported to experience a strong preference for sweet taste and sugar-sweetened foods and beverages [6,7,50]. This could feasibly predispose users of opioids to replace nutritious foods with high-sugar palatable foods that low in nutrients (so-called “empty calories”), and thereby lead to a low dietary intake of many essential micronutrients—including folate.

Although we argue that insufficient intake from food is likely an important cause of poor serum folate status in this population, other factors could be of importance as well. Some pharmaceuticals, such as barbiturates and certain anti-convulsants, are known to impair folate status [1,13]. Due to the high prevalence of somatic and psychiatric health complaints in this population, the use of such medications could be of importance [25,51]. Unfortunately, we were not able to account for this in our study. Chronic HCV infection and other hepatic diseases are prevalent in this population and could, feasibly, impact folate homeostasis through impaired enterohepatic circulation and reduced decreased hepatic storage capacity [52,53]. We did, however, not find any support for this in our analyses.

## 5. Strengths and Limitations

A major strength of this study is its relatively large sample size among a “hard-to-reach” population, and its longitudinal design. Each participant, on average, had over three serum folate measurements over the course of three years. This allowed us to follow patterns in serum folate and predictor variables over time, adding strength to the descriptive analyses and associations presented in this study. This is particularly important as serum folate is highly impacted by very recent intake of food folates, and one single measurement is often not representative true folate status. One important limitation of this study is related to the timing of health assessments and blood samplings; although health assessments were performed annually, the blood samples were not necessarily drawn in exact concurrence with these assessments. Another limitation of this study includes the lack of healthy controls and limited pre-existing knowledge on the serum folate status of the general Norwegian population. This restricts our ability to compare the folate status of this group to that of the general population. Future studies should aim to recruit healthy controls in a case-control manner, or cohort data including both the general population and people with substance use disorder. This will better allow for comparisons between groups. Moreover, most participants in our study cohort were enrolled in OAT for opioid dependence, and although many other substance classes were commonly used, our results may not be generalizable to different SUDs such as predominantly alcohol addictions.

## 6. Conclusions

The folate status of this population of outpatients with SUD was poor. Almost half of the population had serum folate levels below the cut-off that warrants interventions such as nutritional guidance or folic acid supplementation, and a quarter were deficient. Poor folate status was particularly prevalent among those with a more frequent use of multiple substances, those injecting substances, and those prescribed higher dosages of OAT medication, or full agonist OAT. These novel findings shed light on one of the many nutritional challenges faced by people with SUD, especially by those with the most severe substance use. Efforts should be made to address poor micronutrient status, as well as other aspects of suboptimal diet and nutrition, among people with severe SUD.

## Figures and Tables

**Figure 1 ijerph-19-05754-f001:**
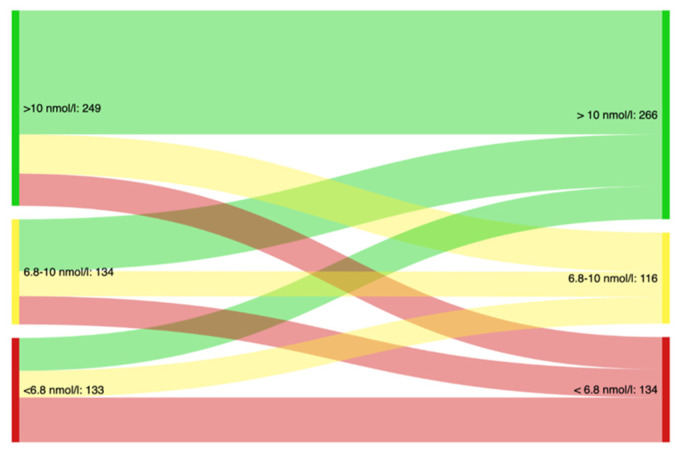
Movement between folate status categories from the first to the second serum folate assessment.

**Figure 2 ijerph-19-05754-f002:**
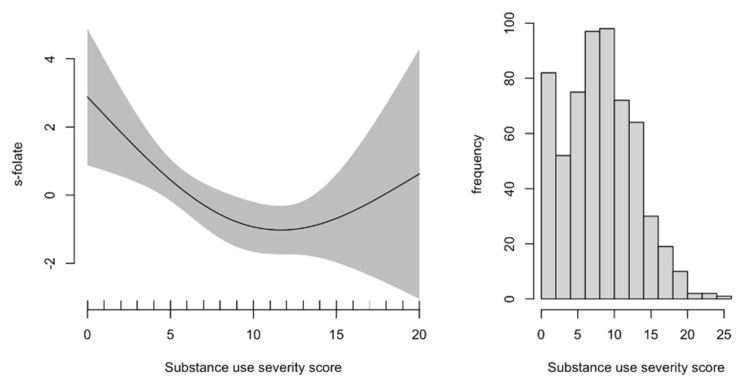
Generalized additive model plot of the association of serum folate concentration with the substance use severity score.

**Table 1 ijerph-19-05754-t001:** Descriptive characteristics of the study population, median serum folate and the prevalence of inadequate folate status and folate deficiency at baseline.

Characteristic	N (%)	S-Folate (nmol/L) Median (IQR ^1^)	Low S-Folate % of Group (CI ^2^)	Deficient S-Folate % of Group (CI ^2^)
**Gender**				
Male	466 (70)	13 (9.6)	48 (43–52)	22 (20–32)
Female	197 (30)	15 (11)	48 (41–55)	26 (18–25)
**Age group**				
<30 years	78 (12)	9.1 (7.1)	59 (48–69)	24 (16–35)
30–39 years	188 (28)	10 (9.3)	50 (43–57)	27 (21–33)
40–49 years	202 (30)	10 (12)	48 (41–54)	22 (17–28)
50–59 years	155 (23)	12 (13)	45 (38–53)	21 (15–28)
≥60 years	40 (6)	14 (12)	25 (14–40)	15 (8–34)
**Education level**				
Primary school	39 (6)	11 (13)	49 (34–64)	28 (17–44)
Middle school	297 (45)	10 (10)	50 (44–54)	26 (21–31)
High school	263 (40)	11 (10)	46 (40–52)	21 (16–26)
≤3 years higher education	51 (8)	13 (14)	43 (30–57)	20 (11–32)
>3 years higher education	13 (2)	13 (19)	46 (23–71)	0 (0–23)
**Housing conditions** ^3^				
Unstable	81 (12)	10 (11)	49 (39–60)	20 (13–29)
Stable	582 (88)	10 (8.3)	47 (43–51)	23 (20–27)
**HCV infection** ^4^	311 (47)	11 (10)	45 (40–51)	23 (19–28)
**Injecting substances** ^5^	318 (48)	9.2 (8.0)	55 (50–60)	24 (20–29)
**Opioid agonist therapy**				
Buprenorphine	352 (53)	11 (11)	44 (39–49)	18 (14–22)
Methadone	229 (35)	9.1 (11)	54 (48–60)	32 (26–38)
Not in OAT	73 (11)	11 (11)	45 (34–57)	15 (9–25)
**Weekly substance use** ^6^				
Alcohol	148 (25)	13 (11)	32 (25–40)	12 (7–18)
Cannabis	297 (49)	9.8 (10)	52 (46–57)	28 (23–33)
Stimulants ^7^	157 (26)	9.4 (6.9)	54 (46–62)	21 (15–28)
Benzodiazepines	230 (38)	9.6 (9.1)	52 (45–58)	26 (21–32)
Non-OAT opioids	85 (14)	11 (10)	47 (37–58)	17 (10–26)
No weekly substance use	144 (24)	11 (12)	44 (37–53)	23 (17–30)
**Overall**	663 (100)	10 (11)	48 (44–51)	23 (20–26)

^1^ IQR, interquartile range, ^2^ CI, 95% confidence interval, ^3^ Stable housing included living in owned or rented housing or at an institution, unstable housing included homelessness, living at temporary camping sites or with friends or family, ^4^ Hepatitis C virus infection, defined as non-zero values on a quantitative HCV-RNA assay at baseline, ^5^ Self-reported injection of any substance during the 12 months prior to the first health assessment, ^6^ Self-reported substance use on a minimum weekly basis during the 12 months. ^7^ Amphetamine, methamphetamine and cocaine.

**Table 2 ijerph-19-05754-t002:** Linear mixed model of serum folate concentration (nmol/L) adjusted for sociodemographic and clinical factors, including substance use patterns.

Fixed Effects	Partly Adjusted ^1^		Adjusted	
	Effect Estimate	Time Trend (per Year)	Effect Estimate	Time Trend (per Year)
	Estimate (CI)	Slope (CI)	Estimate (CI)	Slope (CI)
**s-folate**			*12.8 (10.3, 15.3)*	*2.1 (0.79, 3.4)*
**Gender**				
Male			Reference (0.0)	
Female			0.85 (−0.49, 2.2)	
**Age**				
<30			Reference (0.0)	
30–39			0.71 (−1.3, 2.6)	
40–49			1.5 (−0.46, 3.6)	
50–59			*2.4 (0.32, 4.5)*	
≥60			*4.5 (1.5, 7.4)*	
**OAT dose ratio** ^2^	−0.018 (−1.5, 1.5)	−0.71 (−1.7, 0.33)	0.06 (−1.5, 1.6)	*−1.1 (−2.2, −0.024)*
**Injecting substances** ^3^	*−1.6 (−3.1, −0.13)*	−0.93 (−1.9, 0.014)	−1.2 (−2.9, 0.41)	*−1.2 (−2.3, −0.14)*
**Weekly substance use** ^4^				
Alcohol	*1.9 (0.13, 3.6)*	−0.53 (−1.6, 0.57)	*1.9 (0.15, 3.6)*	−0.60 (−1.7, 0.50)
Cannabis	*−2.1 (−3.6, −0.63)*	0.12 (−0.82, 1.1)	*−1.8 (−3.3, −0.25)*	0.14 (−0.84, 1.1)
Non-OAT opioids	0.32 (−1.9, 2.5)	0.16 (−1.3, 1.6)	0.80 (−1.5, 3.1)	0.22 (−1.3, 1.8)
Stimulants ^5^	−0.93 (−2.6, 0.77)	−0.54 (−1.7, 0.61)	−0.29 (−2.2, 1.6)	−0.38 (−1.7, 0.94)
Benzodiazepines	−1.1 (−2.6, 0.45)	0.47 (−0.50, 1.4)	−0.50 (−2.1, 1.2)	0.84 (−0.24, 1.9)

The table displays the results of a linear mixed model (restricted maximum likelihood regression) estimating associations of serum folate concentration (nmol/L) with sociodemographic and clinical predictor variables at baseline (effect estimates), as well as the impact of predictors on changes in serum folate concentrations over time (time trends per year). Significant results are shown in italics. CI, 95% confidence interval; ^1^ Adjusted for gender and age. In the partly adjusted model, age and gender were included as categorical independent variables in the model together with one of the clinical variables (substance use, opioid agonist therapy, and injecting behaviour) separately, as well as interaction between this variable and time (using identity and timepoints as hierarchical group variables). In the adjusted model, age and gender were also included as categorical independent variables together with all the clinical variables (substance use, opioid agonist therapy, and injecting behaviour). Substance use variables, opioid agonist therapy, and injecting use were also included as independent variables in the models with interactions between each of these variables and time (using identity and timepoints as hierarchical group variables); ^2^ The patients’ prescribed daily dose of opioid agonist divided by the WHO mean expected dose (90 mg for methadone, 18 mg for buprenorphine). In this variable, zero represents no prescribed OAT medication; ^3^ Self-reported injection of any substance during the 12 months prior to the first assessment (baseline); ^4^ Self-reported use of a substance at a minimum weekly basis during the 12 months prior to the first assessment; ^5^ Amphetamine, methamphetamine and cocaine.

## Data Availability

The datasets generated and analysed during the current study are not publicly available due to privacy concerns, but are available from the corresponding author on reasonable request.

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
