# Peer review of "Folate Status as a Nutritional Indicator among People with Substance Use Disorder; A Prospective Cohort Study in Norway"

_ijerph, 2022, doi:10.3390/ijerph19095754_

Round 1
Reviewer 1 Report
The paper focuses on an interesting objective: assess serum folate levels among people with substance use disorder and investigate the impact of factors related to substance use severity on folate status in Norwegian population. The theme is well founded through a consistent literature review. The objectives are correctly defined. The methodology is well-designed and is consistent with the objectives of the study. The interpretation and discussion of results is clear, objective, and consistent. The conclusions summarize well the results obtained and are consistent with the work presented. Only a few minor details remain:
- p3 (line 130) - I think it's not "60 years" but ">=60 years"
- p6 (lines 199-201) - Integrate this information into the legend of Fig 1
- p6 (lines 209-211) - Integrate this information into the legend of Fig 2
- Table 2 presents information regarding the fitted linear mixed model. It would be important to explain how the adjustment was made for the variables gender and age. It would also be important to go a little deeper into the interpretation of the results obtained.
I really enjoyed reading your work! It is very clear and well organized! Congratulations!
Reviewer 2 Report
This prospective longitudinal cohort study investigated the serum folate levels among people with substance use disorder (SUD) to understand the factors related to substance use severity on folate status. Please conduct the concerns below.
- Including the outcomes of opioid agonist therapy (OAT) in the current study requires a clear introduction, such as the merits and/or rationale.
- The current study focuses on SUD for opioid addiction. Why?
- In Table 1, there appears to be no difference in s-folate levels between the non-OAT and OAT groups. Please compare it so that we can discuss it.
- The current study's limitation has been identified as a lack of healthy controls. How to overcome this disadvantage appears to be useful in the discussion.
- Using the published report, the value of the obtained serum folate was not compared to another population, such as Asian or Latin American (s).
- Poor folate status in OAT appears to have been mentioned previously. Please include the citation (s).
- The poor micronutrient status in OAT needs the reference(s) to support.
- Novelty was not indicated in the conclusions.
